# Summary of the Current Status of African Swine Fever Vaccine Development in China

**DOI:** 10.3390/vaccines11040762

**Published:** 2023-03-29

**Authors:** Naijun Han, Hailong Qu, Tiangang Xu, Yongxin Hu, Yongqiang Zhang, Shengqiang Ge

**Affiliations:** 1China Animal Health and Epidemiology Center, No. 369 Nanjing Road, Qingdao 266032, China; 2Key Laboratory of Animal Biosafety Risk Prevention and Control (South), Ministry of Agriculture and Rural Affairs of the People’s Republic of China, No. 369 Nanjing Road, Qingdao 266032, China

**Keywords:** African swine fever, live attenuated vaccine, subunit vaccine, live vector vaccine

## Abstract

African swine fever (ASF) is a highly lethal and contagious disease of domestic pigs and wild boars. There is still no credible commercially available vaccine. The only existing one, issued in Vietnam, is actually used in limited quantities in limited areas, for large-scale clinical evaluation. ASF virus is a large complex virus, not inducing full neutralizing antibodies, with multiple genotypes and a lack of comprehensive research on virus infection and immunity. Since it was first reported in China in August 2018, ASF has spread rapidly across the country. To prevent, control, further purify and eradicate ASF, joint scientific and technological research on ASF vaccines has been carried out in China. In the past 4 years (2018–2022), several groups in China have been funded for the research and development of various types of ASF vaccines, achieving marked progress and reaching certain milestones. Here, we have provided a comprehensive and systematic summary of all of the relevant data regarding the current status of the development of ASF vaccines in China to provide a reference for further progress worldwide. At present, the further clinical application of the ASF vaccine still needs a lot of tests and research accumulation.

## 1. Introduction

African swine fever (ASF) is an acute, febrile, highly contagious disease of pigs caused by the infection with African swine fever virus (ASFV). ASF is on the list of notifiable diseases of the World Organization for Animal Health (WOAH) and is classified as a Class I animal disease in the Chinese list of animal pathogenic microorganisms, which refers to diseases that pose a serious threat to humans and animals and require urgent, severe and compulsory measures for their prevention, control and eradication. ASFV can be transmitted over long distances through pork products, swill and other items, which has set the precedent for transcontinental outbreaks. In 2018, ASF appeared suddenly in China and the development of the African swine fever vaccine was initiated [1,2]. In the past 4 years (2018–2022), several groups in China have presided over/participated in the research and development of various types of ASF vaccines. More than 50 vaccine-related articles have been published (Figure 1), with some studies making marked progress and reaching certain milestones. To provide a reference for further progress worldwide, we have provided a comprehensive and systematic summary of all of the relevant data regarding the current status of the development of ASF vaccines in China. 

## 2. Gene-Deleted Live Attenuated Vaccines (LAVs)

After ASF was first reported in China in 2018, the outbreak strains designated in China were 2018/1 [2], pig/HLJ/18 [3] and SY18 [1] and these were isolated by different research groups and used to conduct relevant vaccine studies. Foreign experience in ASF vaccine research has shown that inactivated vaccines have no reliable protection [4]; cell-passage-attenuated vaccine candidates is relatively difficult to determine the appropriate number of passages [5]; natural attenuated vaccine candidates are rare [6,7,8]; subunit vaccines/or vector vaccines have weak and unstable protection, etc.; so, these vaccine development methods are limited by a degree of uncertainty. In recent years, with the continuous development of gene editing technologies, artificial gene-deleted LAVs have become the most promising strategy for ASF vaccine development [9].

Among them, the MGF 360/505 gene (six genes) and CD2v gene have received wide attention from researchers. MGF360 and MGF505, located in the highly variable left terminal genomic region of ASFV, encode products with common structural similarities [10,11,12]. It was shown that MGF360 and MGF505 genes are involved in the inhibition of interferon (IFN) production [13] and are associated with viral virulence. The deletion of these genes can lead to the reduced virulence of ASFV and to the acquisition of gene-deleted live attenuated vaccines [14,15]. CD2v (EP402R) is an ASFV protein with a sequence homologous to the T lymphocyte surface adhesion receptor CD2, detected in the outer layer of the budding virus [16]. The extracellular structural domain causes the hemadsorption phenomenon [17]. The deletion of the CD2v (EP402R) gene highly attenuates the virulence of ASFV in vivo [14]. In China, several independent research groups have conducted knock-out studies on the major genes of ASFV, and three of them have selected the combination of the deleted MGF 360/505 gene (six genes) and CD2v gene (one gene) as vaccine candidate strains for further evaluation [18,19].

### 2.1. Progress of Seven-Gene-(Dual-Gene)-Deleted LAV Research

Research on the HLJ/18-7GD strain has been highly comprehensive and has achieved the most rapid progress (see Table 1 for details). This gene-deleted live attenuated vaccine is based on the ASFV HLJ/18 virus as the backbone with the deletion of seven genes encoding MGF5051R, MGF505-2R, MGF505-3R, MGF360-12L, MGF36013L, MGF360-14L and CD2v. Safety evaluation and protective efficacy studies conducted on more than 1100 specific pathogen-free (SPF) pigs in the laboratory, as well as experiments on 14,487 commercial fattening pigs and 229 breeding sows in the field, have confirmed that the HLJ/18-7GD strain is genetically stable, does not cause disease or become virulent after inoculation and has no adverse effects on the growth, weight gain or mortality of fattening pigs. The survival protection rate for commercial pigs in the field exceeds 80% [19]. Other seven-gene-deleted strains, such as the ASFV CN2018 ΔMGF/ΔCD2v strain (unpublished data) and ASFV SY18 ΔMGF/ΔCD2v [20], have also had animal experiments conducted and have improved to be safe and effective. However, although the three strains have similar deletion genes, the specific knockout sites are not identical, which, together with the differences in purification processes, culture systems and the number of passages, may lead to some differences in the deletion strains and the related test data.

As with ASFV-G-ΔI177L [21] (US vaccine strain), Lv17/WB/Rie1 [7] (European boar oral vaccine strain) and FK-32/13 [22] (Russian cell passage vaccine strain), HLJ/18-7GD [18] (Chinese vaccine strain) is still in the stage of clinical evaluation. The government has concerns about the safety of attenuated vaccines, the occurrence of post-viral shedding and post-vaccination complications and the inadequate protection of immunocompromised pigs [23], and is cautious about the large-scale promotion of an attenuated vaccine.

### 2.2. Research Progress of Other Gene-Deleted LAVs

During the clinical evaluation of the seven-gene-deleted live attenuated strains, research on gene-deleted live attenuated strains of ASF in China has not stopped, with increasing numbers of participating units and continuous updates of research results (Table 2). In addition to the above-mentioned seven-gene-deleted live attenuated strains, other ASFV gene-deleted strains have been constructed in recent years. Genes that can be knocked out include those related to innate immunity, such as MGF100-1R [24], MGF 110-9L [25], MGF 505-2R [26], MGF 505-7R(A528R) [27,28,29], QP509L/QP383R [30], E120R [31], F317 [32] and L7L-L11L [33], genes related to virus transcription, such as MGF360-9L [34], I226R [35], I215L [36,37] and I267 [38], and genes related to virus structure, such as A137R [39]. Strains with deletions of these genes or their combination with previously reported genes may provide new candidates for vaccine evaluation similar to the ASFV-Δ9L/Δ7R strain with the deletion of both MGF360-9L and MGF505-7R genes [40], and the ASFV-GZΔI177LΔCD2vΔMGF with simultaneous deletion of I177L, CD2v and virulence-associated MGF360-12L-MGF360-14L gene clusters [41]. Evaluations of most of the above vaccine candidates are currently limited to small-scale animal experiments (5–10 animals) in the laboratory, mainly to meet publication/patent requirements, and have not yet been studied on a large scale.

## 3. Other Vaccines

The design and production of subunit or live vector vaccines do not involve the manipulation of the ASF live virus, and challenge protection data are not necessary in the submission content of patent applications. This has led to the emergence of a large number of ASF vaccine patents in China and as of December 2022, the number of ASF subunit and live vector vaccine patents publicly reported in China has reached 69 and 43, respectively, exceeding the number of patents for genetically attenuated live vaccines.

Based on previous experience in animal disease vaccine design and improved understanding based on previous research on the ASF subunit and live vector vaccines, Chinese researchers have adopted similar research strategies for the development of the ASF vaccine (Figure 2). For the subunit vaccine, the protein sequences were analyzed using bioinformatics methods [44,45,46,47] or by using monoclonal antibodies [48] to screen the antigenic epitopes. Once identified, the ASFV antigenic proteins were combined with other bacterial/viral proteins or functional proteins [49,50], so that the expressed ASFV antigenic proteins could self-assemble into a certain structure or improve the expression efficiency to improve the immunogenicity of the ASFV proteins. For the live vector vaccines, different vectors [51,52,53,54,55] and multiple gene expressions [56,57,58] were also evaluated.

However, the major challenges to the development of the ASF subunit or live vector vaccines in China remain, with most of the candidates awaiting protective immunity validation and patent authorization. Some patents relate only to the design concept of combining different genes without an in-depth discussion; so, it is impossible to judge their vaccine application potential.

## 4. Conclusions

Several ASF vaccines, all LAVs, that have entered the clinical trial phase have been urgently suspended due to safety concerns. ASF subunit vaccines are not associated with risks such as recombination, virulence reversion or virulence residues, and have unparalleled advantages in safety compared with LAVs; therefore, ASF subunit vaccines are now a focus of current research. However, further technological advances and basic research are required to successfully develop a safe and effective ASF vaccine.

## Figures and Tables

**Figure 1 vaccines-11-00762-f001:**
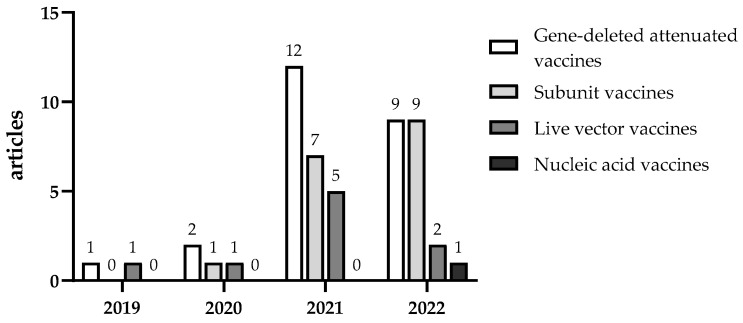
Publication of ASF vaccine articles in China.

**Figure 2 vaccines-11-00762-f002:**
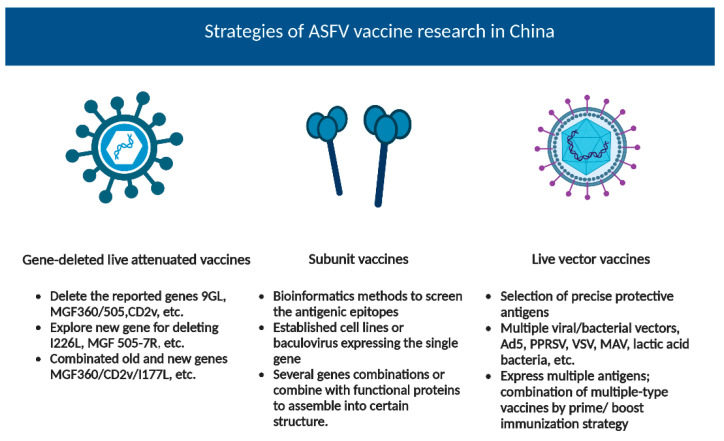
Created using BioRender.com.

**Table 1 vaccines-11-00762-t001:** Advances in ASFV seven-gene-deleted attenuated vaccine (HLJ/18-7GD strain) development.

Period	Progress/Achievements	References
October–November 2018	Isolation and acquisition of endemic ASFV strains in China, completion of whole genome sequence determination and analysis and establishment of infection pathogenesis model	[3]
November 2018–May 2019	Screening out one safe and effective LAV candidate strain HLJ/18-7GD	[18]
December 2019	Approval of the gene-deleted LAV candidate strain for environmental release test after national review	[19]
June 2019–February 2020	Completion of vaccine laboratory product quality research, large-scale production process research and intermediate trial production, as well as biosafety evaluation intermediate test	[19]
March 2020	Approved for clinical trial of veterinary biological products by the Ministry of Agriculture and Rural Affairs	[19]
April–June 2020	Biosafety evaluation environmental release testing and Phase I clinical trial conducted in four independent pig farms	[19]
August–October 2020	Approved by the Ministry of Agriculture and Rural Affairs to enter the biological safety evaluation production test phase, and field biological safety production tests conducted at two enclosed test bases.	[19]
September–October 2020	Approved by the Ministry of Agriculture and Rural Affairs for conducting Phase II clinical trials at four clinical trial enclosed bases	[19]

**Table 2 vaccines-11-00762-t002:** Research progress of ASFV gene-deleted LAV candidate strains in China *.

Year	Targeted Genes	Immunizing Dose	Survival Rate	Challenge Timepoint (dpv)	Challenge Dose	Survival Rate	References
2019	MGF 360/505	10^3^ TCID_50_, 10^4^ TCID_50_	10/10	28	10^3^ TCID_50_	5/5	[20]
	MGF 360/505 + CD2v	10^3^ TCID_50_, 10^4^ TCID_50_	10/10	28	10^3^ TCID_50_	10/10	
2020	MGF 360/505 + CD2v	10^3^ TCID_50_, 10^5^ TCID_50_	8/8	21	200 PLD_50_	8/8	[18]
	MGF 360/505	10^3^ TCID_50_, 10^5^ TCID_50_	8/8	21	200 PLD_50_	8/8	
	9GL + UK	10^3^ TCID_50_, 10^5^ TCID_50_	12/12	21	200 PLD_50_	0/12	
	CD2v	10^3^ TCID_50_, 10^5^ TCID_50_	3/8	21	/	/	
	CD2v + UK	10^3^ TCID_50_, 10^5^ TCID_50_	4/8	21	/	/	
	CD2v + UK	10^4^ TCID_50_	5/5	28	10^4^ TCID_50_	5/5	[42]
2021	I226R	10^4^ TCID_50_, 10^7^ TCID_50_	10/10	21	10^2.5^ TCID_50_/10^4^ TCID_50_	10/10	[35]
	MGF 505-7R	10 HAD_50_	6/6	21	/	/	[27,28]
	MGF 110-9L	10 HAD_50_	3/5	17	/	/	[25]
	QP509L/QP383R	10^4^ HAD_50_	6/6	17	10^2^ HAD_50_	0/6	[30]
2022	MGF360-9L	1HAD_50_	4/5	17	/	/	[34]
	I267	10 HAD_50_	1/6	21	/	/	[38]
	MGF360-9L + MGF505-7R	10^4^ HAD_50_	6/6	23	10^2^ HAD_50_	5/6	[40]
	EP153R/EP402R + MGF 360-12L/13L/14L	10^5^ TCID_50_	5/5	28	10^2^ HAD_50_	5/5	[43]

* Gene-deleted strains with reduced virulence after deletion; single-dose immunization and intramuscular route are primarily used.

## Data Availability

Not applicable.

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
