# Peer review of "Summary of the Current Status of African Swine Fever Vaccine Development in China"

_vaccines, 2023, doi:10.3390/vaccines11040762_

Round 1

Reviewer 1 Report

In my opinion China made a great step forward in last 3 years on the field of ASF research and vaccine development. And presented overview of current status of ASF vaccine development and perspectives of vaccine application, written by Chinese scientist are very important for research community.

I have couple recommendations to the authors:

- almost in every paper on ASF vaccine authors make a statement that "There is still no commercially available vaccine" but it is publically available information that in June 2022, the Ministry of Agriculture and Rural Development of Vietnam issued a commercial circulation license for the vaccine against African swine fever, researched and made by the Central Veterinary Medicine Company (Navetco). Formally this is the first commercial ASF vaccine.

Concerning the Table 2. Research Progress of ASFV Gene-deleted Live Vaccine in China. I recommend to add the information about the route of vaccine admission (intramuscularly, intranasal), the period between immunization and challenge (21 days, 28 days) and information about the scheme of immunization (single/boost). I think it is significant information about tested vaccine candidates.

Author Response

Dear Reviewer

Thanks for your comments and suggestions. Please find our responses below.

1. Almost in every paper on ASF vaccine authors make a statement that "There is still no commercially available vaccine" but it is publicly available information that in June 2022, the Ministry of Agriculture and Rural Development of Vietnam issued a commercial circulation license for the vaccine against African swine fever, researched and made by the Central Veterinary Medicine Company (Navetco). Formally this is the first commercial ASF vaccine.

R: We’d like to descript it as "There is still no credible commercially vaccine".   Though we can call it the first commercial vaccine, it was actually used in limited quantities in limited areas, and some problems have been reported since. Technically, it's only in the evaluation stage of large-scale clinical trials.

2. Concerning the Table 2. Research Progress of ASFV Gene-deleted Live Vaccine in China. I recommend to add the information about the route of vaccine admission (intramuscularly, intranasal), the period between immunization and challenge (21 days, 28 days) and information about the scheme of immunization (single/boost). I think it is significant information about tested vaccine candidates.

R:The relevant information has been supplemented, please check the manuscript.

Reviewer 2 Report

The manuscript “Summary of the current status of African swine fever vaccine development in China” summarizes the efforts in the last four years since 2018 to develop a safe and protective vaccine against African swine fever (ASF). Various types of ASF vaccines are under development so far. The different approaches are either based on gene-deleted live attenuated vaccines (LAV), subunit or live-vector vaccines. The article is informative by giving the status quo of ASF vaccine development. These efforts are important to know for several countries.

I have some minor suggestions for adaptation of the article:

1.    At several places in the text spaces are lacking between words, so that sentences are hard to read. The whole text should be checked in detail for these formal mistakes (e.g.: line 12 “lethalandcontagious”, line 50 “vaccinesdo”, line 82 “(HLJ/18-7GD strain)development”, line 119 “onprevious”)

2.    Line 31-43: provide some references within this paragraph, e.g. for the first description of the virus in China.

3.    Line 47-48: Why were 3 different virus strain in parallel introduced to China ? Is there a hypothesis on routes of introduction and origins of the different strains ?

4.    Line 52-53: give reference for this statement

5.    Line 53-55 give reference for this statements.

6.    Line 57-58 please give short explanation about the function and meaning of MGF and CD2v genes. Shouldn´t the gene names be in italics and small letters ?

7.    Line 59: give reference

8.    Line 61: can you give short description about characteristics of the HLJ/18-7GD strain ?

9.    Line 69 sentence should be rephrased: ”Other seven-gene (…) have also conduct (…)” What is meant ?

1 Line 69: reference 6 refers to only one vaccine strain, but in the text is written about three strains.

1 Line 73: give reference

1 Table 2 on page 4: please add the route of inoculation in third column and of the challenge dose in 5th column.

1 Legend to table 2: I cannot find a and b superscript

1Line 113: Why are challenge protection data not necessary for other than LA vaccines ? Please explain.

1Line 137: “gene-deleted LIVE attenuated vaccines”

1 Line 149 Vaccine candidates must prove their protectivity although protectivity challenge tests are not necessary as stated above ? Please explain (are vaccine candidates immediately used under field conditions ?)

1 Line 154-155: the safety concerns have not been adequately addressed in the text. Please add in the text (in paragraph line 61-73).
